# Electrophysiological responses to conspecific odorants in *Xenopus laevis* show potential for chemical signaling

Heather J. Rhodes[ID]1,2*, Melanie Amo1,2

1 Department of Biology, Denison University, Granville, Ohio, United States of America, 2 The Grass Lab, Marine Biological Laboratories, Woods Hole, Massachusetts, United States of America

* rhodesh@denison.edu

## Abstract

The fully aquatic African clawed frog, *Xenopus laevis*, has an unusual and highly adapted nose that allows it to separately sample both airborne and waterborne stimuli. The function of the adult water nose has received little study, despite the fact that it is quite likely to receive information about conspecifics through secretions released into the water and could aid the frog in making decisions about social and reproductive behaviors. To assess the potential for chemical communication in this species, we developed an *in situ* electroolfacto-gram preparation and tested the olfactory responses of adult males to cloacal fluids and skin secretions from male and female conspecifics. We found robust olfactory responses to all conspecific stimuli, with greatest sensitivity to female cloacal fluids. These results open the door to further testing to identify compounds within cloacal fluids and skin secretions that are driving these responses and examine behavioral responses to those compounds. Understanding the role of chemical communication in social and reproductive behaviors may add to our rich understanding of vocal communication to create a more complete picture of social behavior in this species.

**Data Availability Statement:** Data are available in the Denison Digital Commons https://digitalcommons.denison.edu/ (more specific URL to be provided after acceptance).

## Introduction

*Xenopus* and other *pipid* frogs are fully aquatic species that spend their adult lives in ponds rather than becoming terrestrial as adults. Although their anuran ancestors lived their adults lives out of the water, these species have adapted to aquatic life with numerous specializations over the last 140 million years or more [1, 2]. These include specializations to the olfactory system to allow adult animals to separately sample both airborne and waterborne stimuli [3–7]. Adult *X. laevis* have two chambers within the nose with a valve at the external naris that allows either the air nose to be open when above water, or the water nose to be open below the surface (Fig 1A). The air nose, or principal cavity, connects to the respiratory tract and contains an olfactory epithelium similar to that seen in all adult anurans; it may be used to find new ponds during overland migration [3–6]. Also similar to other anurans, *X. laevis* have a vomeronasal organ at the base of the principal cavity and adjacent to the choana (the opening that connects the oral cavity with the principal nasal cavity) which likely samples waterborne chemicals

**Funding:** Funding was provided by The Grass Foundation https://grassfoundation.org/ (HJR), and Denison University https://denison.edu/, including the R.C. Good Faculty Fellowship at Denison University (HJR), and the Helen L. Yeakel Summer Research Fund at Denison University (MA). The funders had no role in study design, data collection and analysis, decision to publish, or preparation of the manuscript.

**Competing interests:** The authors have declared that no competing interests exist.

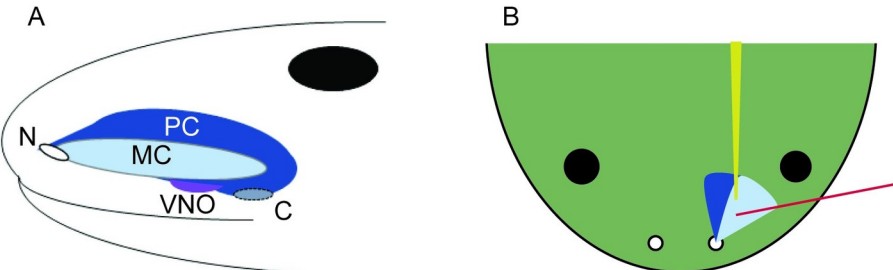

**Fig 1. Location of the water nose and electrode placement.** The medial cavity (MC, light blue) or water nose is a blind-ended compartment, opening only at the naris (N). The principal cavity (PC, dark blue) or air nose is a separate compartment that connects the naris to the oral cavity via the choana (C). The vomeronasal organ (VNO, purple) sits in the inferior aspect of the PC. Large black dots represent the animal's eyes. (A) Schematic of olfactory anatomy, lateral view, as described by Reiss & Eisthen, 2014 [5]. (B) Schematic of experimental preparation, dorsal view. The EOG recording electrode is shown in red; it was placed along the medial wall of the MC. The profusion system that was used to deliver stimuli is represented in yellow.

originating from the nasolacrimal duct or the choana [4–6]. The water nose is a dead-end chamber, often referred to as the medial cavity (despite it being lateral and ventral to the principal cavity), through which water is actively circulated when the animal is submerged due to the pulsation of the lateral nasal wall [8, 9]. The water nose contains a separate olfactory epithelium that resembles the larval epithelium of anurans, showing specializations for waterborne odorants, including a mix of ciliated and microvillous receptor neurons expressing OR1 (class I), OR2, and V1R receptors [3–6, 10–12]. The water nose may be important for finding food, alerting to predators, and acquiring information about conspecifics via chemical cues in the water [8, 10, 12–14].

*X. laevis* social and reproductive interactions have been well studied [15–19] but significant gaps remain in our understanding of how sensory or hormonal cues lead to particular behaviors. There is no evidence that these animals are territorial; instead a population will share space within a pond where they have a prolonged breeding period, with females entering sexual receptivity asynchronously during the rainy season [20, 21]. These animals use their extensive vocal repertoire for social and reproductive communication, with males and females calling to each other, as well as male-male vocal interactions [16, 17, 21]. Males produce more advertisement calls when a female is present, but it is unclear what sensory cues drive the increased calling. When males are housed together, they do not chorus; in fact, certain males tend to do most of the calling, suggesting a social hierarchy [22, 23]. This may involve an assessment of self (endocrine state, for example) relative to others (perhaps using body size or condition, calling, or chemical cues). Males also select among different reproductive tactics [24], which may depend on a similar assessment of conspecifics.

Chemosensory signaling may be an important missing piece of this puzzle. We do not know what sensory cues prompt different vocalizations, particularly for males; nor do we know what causes males to be dominant or subordinate in vocal or clasping interactions. Looking at other species, it seems plausible that chemical communication could play an important role in these social interactions by allowing animals to learn about nearby conspecifics. Chemical signals are frequently used for social and reproductive signaling across all taxa, including amphibians [25–29] and other aquatic vertebrates, such as fish [30]. While vocal communication has traditionally received more attention in anurans, cases of chemical communication have been documented [13, 14, 31–33]. Previous behavioral or physiological studies of *X. laevis* chemosensation have yet to address these questions, largely focusing on responses to food stimuli [8, 10] and on larval olfactory physiology [34–37].

To assess the role of waterborne odorants in adult *X. laevis* social interactions, we developed the first *in situ* electrolfactogram (EOG) preparation for this species, allowing us to record receptor potentials in the water nose. We then used our EOG preparation to test whether male *X. laevis* could detect cloacal fluids or skin secretions from male and female conspecifics and determined the sensitivity of the nose to these potential social stimuli. Cloacal fluids consist primarily of urine but may also contain chemicals from reproductive or gastrointestinal tracts, given the confluence of these systems in the cloaca. Urine, which contains hormones, hormone metabolites, bile acids, and other species-specific, sex-specific, and condition-dependent molecules, is a common source of social chemical signals in other species [30, 37–41]. Amphibian skin secretions also contain a variable mix of chemicals including a variety of peptides, proteins, antimicrobial substances, and toxins [42]. Several pheromones have been identified in the skin secretions of other amphibians [13, 14, 27–29, 31, 32].

## Methods

### Animal handling and *in situ* olfactory preparation

All animal handling and experiments were conducted with the oversight and approval of the Institutional Animal Care and Use Committee at the Marine Biological Laboratory (MBL), Woods Hole, Massachusetts (protocol number 17-07H-Final) as well as the approval of the Denison University Institutional Animal Care and Use Committee.

All animals were sexually mature adults procured from and housed in the National Xenopus Resource (NXR) at the MBL. Frogs were housed in same sex tanks at 18–20 deg C with a 12:12 light cycle. Frogs were fed a 1:1 mix of adult frog brittle (Nasco) and Bio-Trout pellets (Bio-Oregon).

A total of 27 male wild type *Xenopus laevis* (7.9 ± 0.5 cm snout-vent length; 64.0 ± 9.9 g) were utilized for this study, with the first 17 animals used to understand nasal anatomy, establish recording procedures, and pilot a range of potential stimuli. The final 10 animals were used to collect the data presented here. Additionally, several adult male and female *Xenopus laevis* belonging to the NXR were handled briefly and with permission from the NRX to collect stimuli as described below. No physiological recordings were made in female animals for this study. Experiments were conducted during the summer of 2017 (June–August) at the MBL.

The male animals used for physiological recordings were anesthetized with MS222 dissolved in phosphate buffered saline; dosing was 0.15 mg/g body weight, injected subcutaneously into the dorsal lymph sac. Once the frog was deeply anesthetized, we placed it on ice for 5 to 10 minutes before euthanizing by double pithing the frog [43]. By destroying the central nervous system, pithing achieved euthanasia, including terminating all motor activity while leaving the olfactory epithelium intact and functional. This created an *in situ* preparation for testing olfactory responses. The frog's body was placed in a custom chamber that elevated its naris above the rest of its body. Ice was placed over the frog's body to help maintain healthy tissues for as long as possible by lowering the metabolic rate.

We opened the frog naris with small surgical scissors, removing superficial tissue and underlying cartilage until we exposed the medial olfactory cavity (the water nose; Fig 1). The water nose cavity was continuously perfused with room temperature saline at 2–3 ml/min using a gravity fed system to keep the tissue moist and provide a path for stimulus delivery and wash out. Excess fluid flowed freely out of the cavity and ultimately passed through a drain at the bottom of the frog chamber and into a waste collection. Saline was selected to mimic ionic concentrations in olfactory mucosa [44, 45] and consisted of 55 mM NaCl, 10 mM KCl, and 4 mM $CaCl_2$ in Millipore purified deionized water, brought to pH of 7.5 using NaOH.

## EOG recording

Electroolfactogram (EOG) recordings were made using a silver/silver-chloride electrode placed in a glass pipette, tip diameter ~100 μm, tip-filled with 1% agar (Sigma A1296, dissolved in saline) and backfilled with saline. The EOG electrode and reference electrode were connected to a head stage and amplifier (AM Systems 3000) for differential recording. The amplifier was set to DC (no high pass filtering) with notch filtering on, low pass filter at 1 KHz, and gain at 1000x. Data was digitized with a Digidata Micro 1401 (CED, Cambridge, UK), and continuously recorded at a rate of 10 kHz with Spike2 software (CED).

The EOG electrode was held by a micromanipulator and placed such that the tip was submerged in perfused saline, just above the olfactory epithelium along the medial wall of the water nose based on visual landmarks (Fig 1). A reference electrode was placed in the mouth. Electrode position and prep viability were assessed by delivering positive and negative control stimuli (methionine and saline, see stimuli below). If we did not record a normal EOG signal for methionine (a characteristic negative deflection lasting 2–3 seconds with expected latency based on the perfusion and stimulus deliver system described below), we would wash out the stimulus, reposition the recording electrode or change the glass pipette, then try again. Once a recording site was established, a range of stimuli were tested.

## Stimulus acquisition and delivery

Stimuli consisted of male and female cloacal fluids, male and female skin secretions, and several positive and negative controls. Amino acids (1mM L-methionine (Sigma M5308) dissolved in saline and 1 mM L-alanine (Sigma A7469) dissolved in saline) were used as positive controls because they are reliably detected chemical signals in this and other species [10, 11, 25, 34, 35, 46–50]. Negative controls included saline controls (saline identical to that being perfused was injected into the perfusion line to control for mechanosensory response to changes in flow rate or pressure), and cloacal- and skin-specific controls (described below; these controlled for contaminating odorants).

To collect cloacal fluids, we gently held an adult frog and placed a small piece of new, clean polyethylene tubing inside the cloaca of the frog and waited for fluid to move down the tube by capillary action. Not all frogs yielded fluid, those that did typically yielded 10–100 μl. We performed this process with both male and female frogs, until we had successfully collected fluids from several frogs of each sex. All animals were sexually mature adults, but specific hormonal states or reproductive histories were not known. Samples were pooled by sex, creating cocktails of male or female cloacal fluids.

Skin secretions were collected using the hassle bag technique [51]. We rinsed tank water from each frog by gently spraying it with deionized water, then placed each frog into a new plastic sandwich bag (made of low-density polyethylene) and massaged the frog gently for approximately 1 minute to stimulate mucosal secretion. Frogs were then returned to their home tanks and the contents of each bag was collected into a microcentrifuge tube. Again, we sampled several male and female frogs; samples were pooled by sex, creating cocktails of male or female skin secretions.

Because our collection techniques could introduce non-biological odorants into our samples (from plastics, for instance), we created cloacal- and skin-specific control stimuli. To do so, we passed saline through all the steps of collection (including either polyethylene tubing or sandwich bags, and microcentrifuge tubes) for each process.

Samples of all stimuli were aliquoted and frozen to ensure consistency across experiments. Freshly-collected and frozen stimuli were compared during pilot experiments, and no difference in response was observed. Saline used for perfusion was made fresh for each experiment.

Because cloacal fluids were difficult to collect and were collected in small volumes, they were diluted 1:100 in saline before being aliquoted and frozen. Skin secretions were diluted 1:10 before freezing. At the start of each experiment, a set of stimuli was taken from the freezer, defrosted, and serial dilutions were performed using the freshly made saline to achieve a range of dilutions.

To deliver a stimulus during an experiment, 50 μl of the stimulus was injected into a port in the perfusion line carrying saline to the olfactory epithelium. After injection, the stimulus reached the olfactory epithelium after several seconds and somewhat diluted. To better understand the time delay and dilution of stimuli, we calibrated the system without an animal present. Specifically, we ran de-ionized water through the perfusion system and injected 50 μl of 1M NaCl in place of a stimulus. Samples were collected from the perfusion system output and tested on an osmometer. Most of the salt was detected between 5 and 10 seconds after injection, with approximately 1:5 dilution of the peak salt concentration. Actual, instantaneous concentrations experienced by the olfactory epithelium could vary from this estimate due to the way liquid flowed across the epithelium (subtle pooling or mixing could alter the concentration over time) or differences in the temporal resolution of our sampling vs. the temporal resolution of the olfactory epithelium (where long, slow receptor potentials suggest integration over time). Stimulus wash out was also characterized: the measured salt concentration was reduced to less than 1% 20 seconds after injection and undetectable after 30 seconds.

Once a good EOG recording site was established for an animal, stimuli were run in blocks, with each block testing one type of stimulus (e.g., female cloacal fluid) at different dilutions. All blocks began with a positive control (typically 1 mM methionine, but when methionine was the test stimulus, alanine was used as the positive control), followed by a wash (for the wash, ~0.5 ml saline was slowly injected into the perfusion line over several seconds to ensure the injection port was clear of stimuli), then a saline stimulus (50 μl) was run to ensure the wash was complete and no EOG signal was detected (Fig 2). Next, we began the test stimulus at its lowest concentration, followed by a wash and a saline control as before (Fig 2). Then we would deliver the second most dilute test stimulus, followed by washes and controls, and so on, until we reached the least dilute stimulus. If the test stimulus was either skin secretions or cloacal fluids, we also ran the appropriate stimulus-specific control in between each test stimulus. After completing the stimulus sequence from most to least dilute, we would repeat it again

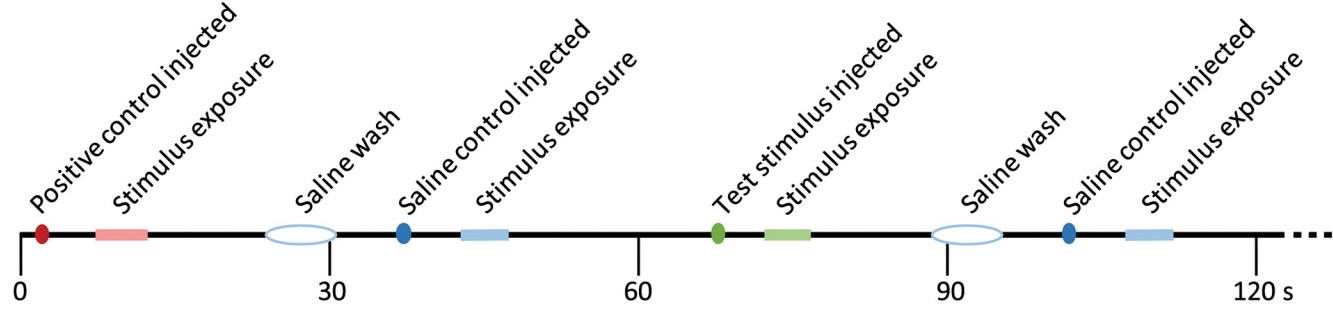

**Fig 2. Sample experiment timeline.** This timeline represents a short segment of an experiment, illustrating the order and timing of events. Stimuli were injected into a port in the perfusion line (colored dots), traveled down the perfusion line for several seconds, then washed across the olfactory epithelium ("stimulus exposure" period, marked on timeline). EOG responses were observed during this exposure time if evoked by the stimulus. Positive controls (shown in red; amino acids such as methionine and alanine) were used to elicit a strong EOG response at the start and end (not shown) of each recording block. After positive controls or test stimuli, a saline wash was administered to ensure the port and line were clean (white oval). Saline was also used as a negative control (blue). Test stimuli (green) at various dilutions were alternated with negative controls for the bulk of each recording block.

in the opposite order–least to most dilute–with appropriate washes and controls in between. This was done to ensure there was no order effect, and indeed we saw no systematic difference in EOG amplitude when we compared low-to-high vs. high-to-low sequences. The stimulus block was ended by repeating the positive control stimulus, followed by a wash. The time between any two stimuli was not automated and thus varied slightly but was almost always >30 seconds, and often closer to 60 seconds. We saw no signs of adaptation in our data; response amplitudes were similar regardless of stimulus order and even when the same stimulus was repeated several times with washes and controls in between. Additional blocks were then run to examine responses to other kinds of stimuli, as long as good quality recordings could be collected. Not all types of stimuli were run successfully in all preparations.

## Data analyses

To quantify EOG responses for dose response data, we measured the amplitude of the EOG signal and calculated z-scores for the amplitudes of signals in response to test stimuli at each dilution relative to control stimuli.

EOG amplitudes were determined by taking the minimum value from the EOG trough and subtracting the average baseline value taken from a 1 second range, starting 2 seconds prior to EOG onset. Because we recorded using a DC amplifier and without low frequency filtering, the baseline showed significant drift at times (Fig 3). Thus, we performed the following baseline corrections: An average post-stimulus baseline was measured from a 1 second range shortly after the end of EOG signal (the time period was set based on the timing of the positive control stimuli at the start and end of each block; this was typically 10 or more seconds after the pre-stimulus baseline). If the pre and post stimulus baselines differed by 0.1 mV or more, the slope of the baseline was calculated. An adjusted baseline value at the time of the EOG trough was then calculated using the slope and the time of the EOG minimum; the trough value was subtracted from the adjusted baseline to determine EOG amplitude.

For each block of stimuli, the average EOG amplitude was calculated for each dilution of the test stimulus (typically there were two trials of each dilution per block). The average and standard deviation of the EOG amplitude for the stimulus-specific negative control were also calculated (typically there were 12 trials of the control per block). Z-scores were then calculated according to the following formula, where $X_i$ is the average EOG amplitude to a test stimulus, $\mu$ is the average EOG amplitude to control, and $\sigma$ is the standard deviation of the EOG amplitude to control:

$$z = \frac{x_i - \mu}{\sigma}$$

Without behavioral data, we cannot know the detection threshold for stimuli. Thus, a threshold of $z \geq 2$ was set to represent responses that are likely detectable as different from control, since such responses would be greater in amplitude than 97.7% of control responses.

## Results

We successfully recorded EOG responses to biologically relevant stimuli in adult male *Xenopus laevis*. We saw large and reliable EOG responses to the amino acids methionine and alanine (Fig 3A) as well as to conspecific cloacal fluids (Fig 3B and 3C) and skin secretions.

We tested the dose dependence of the responses to methionine and found responses declined in amplitude with each 10-fold dilution (Fig 4). For each animal, the average EOG amplitude for a given stimulus type was converted to a z-score using the average and standard deviation of EOG amplitude for control stimuli (for amino acids, saline was used as a control;

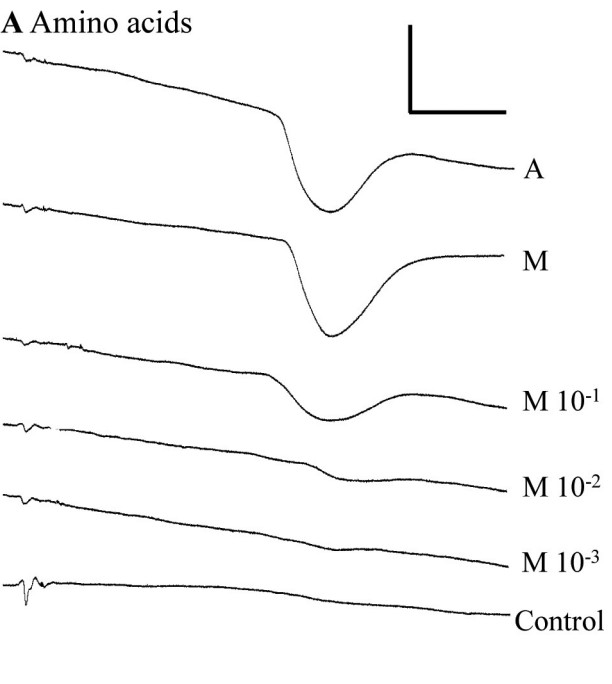

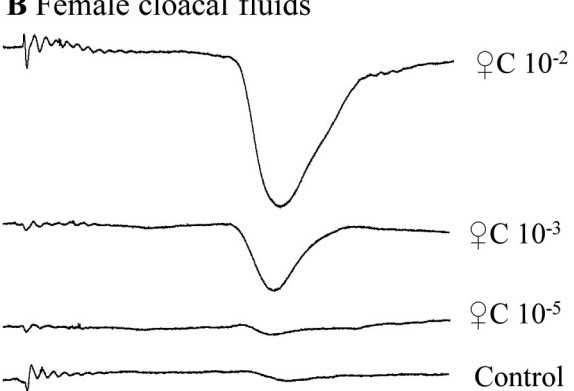

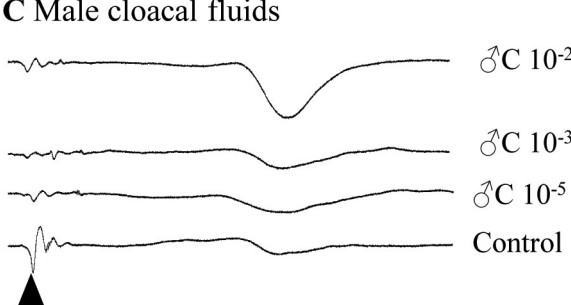

**Fig 3. Sample EOG recordings.** EOG traces are shown in response to (A) amino acids, (B) female cloacal fluids, and (C) male cloacal fluids. The stimulus was injected into the perfusion line at the beginning of each trace (indicated with the black caret), creating a small stimulus artifact, with EOG response occurring 6–8 seconds later (large downward deflections). Stimuli and relative concentration are indicated to the right of each trace (A is 1 mM alanine; M is 1 mM methionine; C is cloacal fluid). Control in (A) was saline; controls in (B) and (C) were cloacal-specific controls (saline passed through the cloacal fluid collection process). Scale bar is 1 mV (vertical) and 3 s (horizontal). Data were from Frog 8; summary data can be found on subsequent figures.

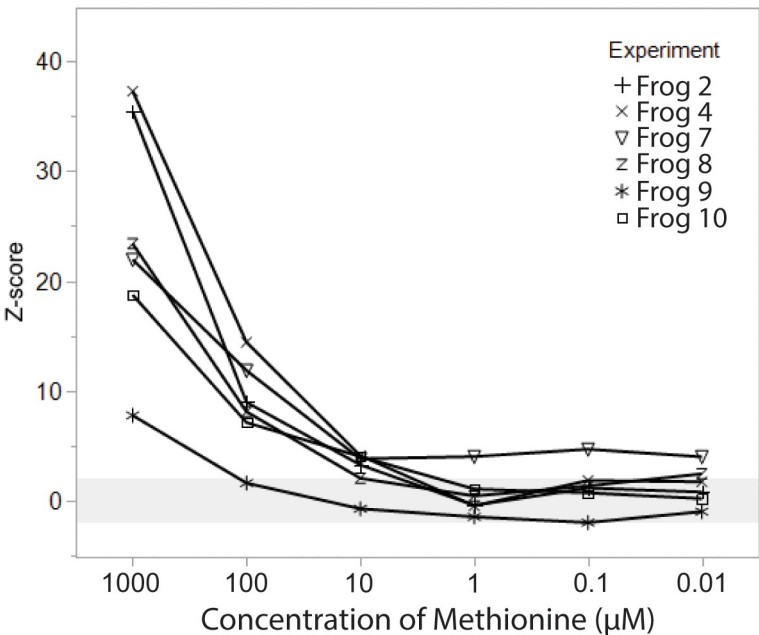

**Fig 4. EOG responses to different concentrations of methionine.** EOG amplitude, represented as z-score relative to control (saline) stimuli, are shown for 6 individuals across 6 concentrations, ranging from 1 mM to 0.01 μM L-methionine. EOG response declined at lower concentrations, with the detectability threshold (z ≥ 2) falling between 10 and 1 μM for most animals. Individual animals are shown with distinct symbols; light gray horizontal band from z = -2 to z = 2 indicates responses that may not be distinguishable from saline control.

for cloacal fluids and skin secretions, saline was run through all equipment used for stimulus collection to create cloacal and skin-specific controls). A stimulus was considered "detected" if the z-score was 2 or greater. The detection threshold for our preparation was between 1 and 10 μM methionine, with 5 of 6 individuals showing detection at 10 μM, and only 1 of 6 showing detection at 1 μM. Note that these and other concentrations were the original concentrations of the stimuli, and we estimate there was an additional 5-fold dilution of stimuli when the stimuli reached the olfactory epithelium.

We found reliable EOG responses to conspecific cloacal fluids which varied in magnitude and detection threshold depending on whether the cloacal fluids were taken from male or female animals (Figs 3 and 5). EOG responses to female cloacal fluids were strong, showing detection in all 7 animals tested for a 1:100 dilution. Detection threshold was between 1:1000 and 1:100,000, with 5 of 7 animals detecting the stimulus at 1:1000 and no animals detecting the stimulus at 1:100,000. The response to male cloacal fluids was less robust. While all animals detected the stimulus at 1:100 dilution, EOG amplitudes were smaller (resulting in smaller z-scores). Only 2 of 6 animals detected male cloacal fluids at 1:1000 dilution and no animals detected it at 1:100,000, suggesting the detection threshold may be close to 1:1000.

Skin secretions from male and female animals also produced robust responses. At 1:10 dilution, all animals showed strong EOG signal, well above control, to skin secretions taken from both male and female animals (Fig 6). At 1:100 dilution, responses decreased but were still detected by 6 out of 7 animals tested with female skin secretions and 4 out of 4 animals tested with male skin secretions. Female skin secretions produced detectable responses in 2 animals at 1:1000 and 1:100,000 dilutions. Male skin secretions evoked just detectable responses in 1 animal at 1:1000 and in a different animal at 1:100,000.

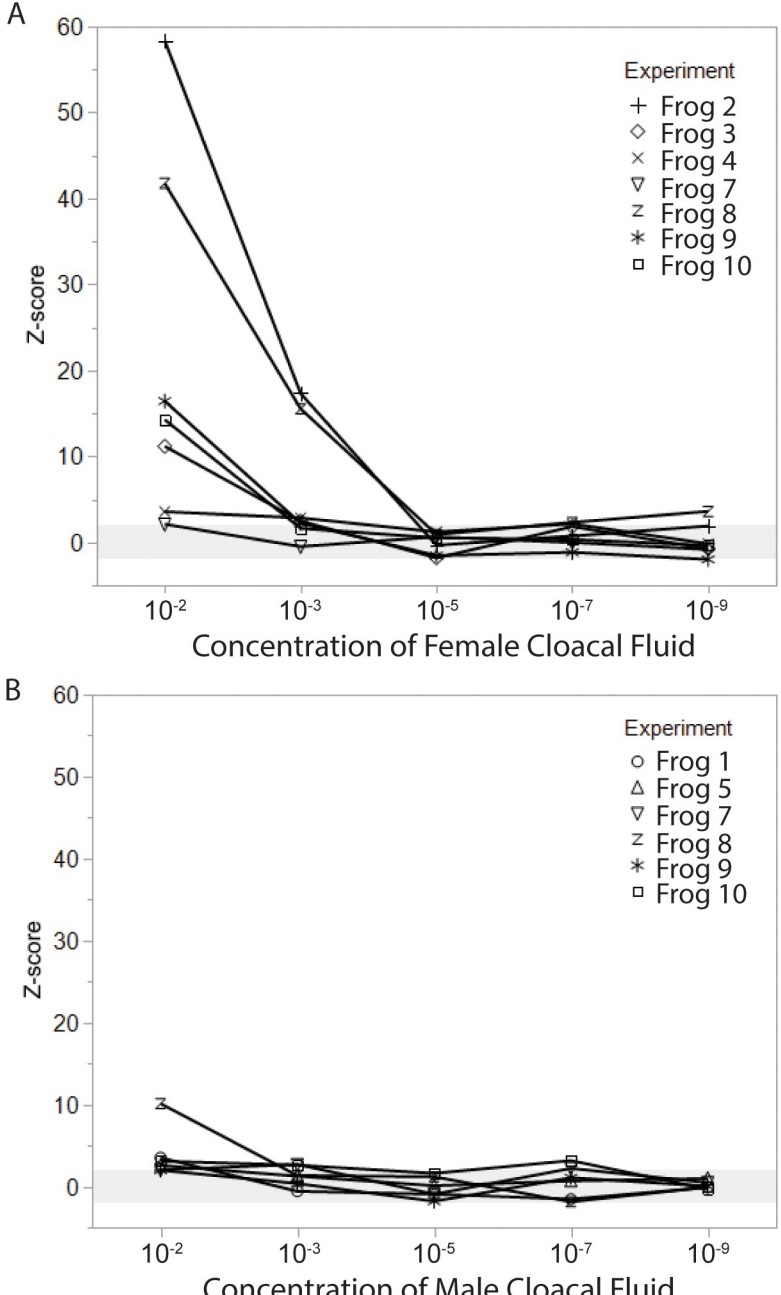

**Fig 5. EOG responses to different concentrations of cloacal fluids from female and male conspecifics, represented as z-score relative to cloacal control stimuli.** (A) EOG amplitudes in response to female cloacal fluids were robust at $10^{-2}$ concentration, and declined at lower concentrations, with the detectability threshold ($z \geq 2$) falling between $10^{-3}$ and $10^{-5}$ for most animals. (B) EOG amplitudes to male cloacal fluids were far smaller, with a detection threshold between $10^{-2}$ and $10^{-3}$ for most animals.

Control stimuli for cloacal fluids and skin secretions consisted of saline passed through the same type of plastics used to collect and store either the cloacal fluids or skin secretions; they were then aliquoted and frozen, just like the test stimuli. These controls often evoked small EOG responses themselves (Fig 3B and 3C), unlike the saline controls used for amino acid stimuli (Fig 3A). This demonstrates the sensitivity of this preparation and the need for careful controls, as even "clean" laboratory equipment may shed odorants that can contaminate stimuli.

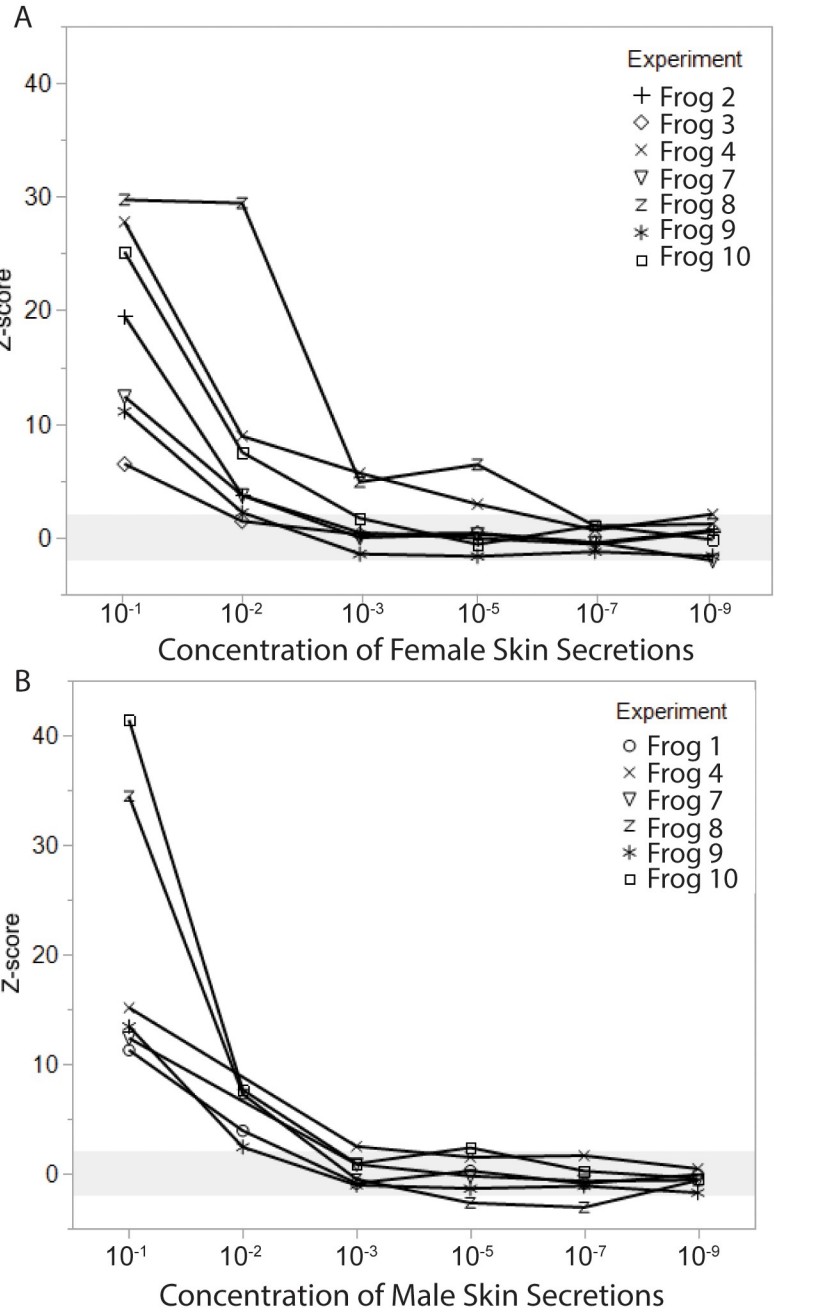

**Fig 6. EOG responses to skin secretions from female and male conspecifics, represented as z-score relative to skin control stimuli.** (A) EOG amplitudes in response to female skin secretions were well above the detection threshold at concentrations of $10^{-1}$ and $10^{-2}$, and may still have been detectable for some animals at $10^{-3}$ and $10^{-5}$. (B) EOG responses to male skin secretions were similar to those seen in A for the most concentrated stimuli, with a detection threshold between $10^{-2}$ and $10^{-3}$ for most animals.

## Discussion

We successfully recorded olfactory responses to conspecific odorants in *X. laevis*, showing that male *X. laevis* likely detect chemicals in female cloacal fluids and in male and female skin

secretions. Our *in situ* EOG preparation worked well, generating responses to amino acid stimuli comparable to other EOG and calcium imaging studies in aquatic amphibians [25, 34, 35, 48, 50]. Using our arbitrary detection threshold of $z \geq 2$, we found reliable responses to the amino acid methionine at concentrations of 10 μM (actual concentration estimated to be closer to 2 μM based on dilution in the stimulus delivery system; see methods). These results are similar to the findings of Breunig and colleagues, which found individual olfactory receptor neurons in larval X. laevis had detection thresholds for methionine ranging from 0.2 to 200 μM [50].

Male *X. laevis* showed robust olfactory responses to conspecific cloacal fluids. Responses to female cloacal fluids showed particular strength and sensitivity, with most animals likely to detect the stimulus at dilutions of 1:1000 or more. Responses to male cloacal fluids were much weaker and just detectable at a dilution of 1:100. This result suggests the presence of a female-specific odorant in cloacal fluids that male *X. laevis* could use to help locate a mate or to determine when advertisement calling would be most advantageous. Female cloacal fluids contain a wealth of potential signal molecules, including hormones and hormone metabolites [37, 38], at least some of which can be detected by olfactory receptors in *X. laevis* tadpoles [37]. In teleost fish, it has been well documented that male fish can detect hormones and hormone metabolites in the urine of female fish, as well as bile acids, and use that information to alter behavior in appropriate ways, such as initiating courtship behaviors in the presence of a reproductively active female [30, 39, 52]. There is evidence that other amphibians may also gain information about the reproductive status of conspecifics or change behavior patterns because of hormones released into the water by conspecifics [41, 53]. Additional testing could elucidate if similar chemical signaling occurs in *Xenopus.*

Olfactory responses to male and female skin secretions were more similar in magnitude and sensitivity, with most animals detecting the stimuli at 1:10 and 1:100 dilutions from either sex. However, two animals appeared to show detectable responses to female skin secretions down to the 1:10,000 dilution, indicating the possibility for greater sensitivity. Skin secretions have been shown to contain pheromones in other species of anuran amphibians; these include signals involved in mate attraction, mate choice, and reproductive competition or aggression [13, 14, 26, 31, 32]. Skin secretions may also contain a variety of antimicrobial peptides and toxins that could be used to identify conspecifics [42]. Given the close proximity of the *X. laevis* male nose to the skin of a conspecific held in amplexus (the reproductive position), there is certainly opportunity to sample odorants released by the skin [24, 28].

Chemosensory information about conspecifics of both sexes could influence male *X. laevis* behavior in important ways. Male *X. laevis* produce different vocalizations [22] and adopt different clasping behaviors [24] depending on the animals they are housed with. Assessing the chemicals released in conspecific cloacal fluids and skin secretions may help males choose the most appropriate and adaptive behavior for its social circumstance. Males may use a combination of chemical and auditory signals to decide when to call and what vocalization to produce. Such multimodal signaling would not be unusual [53–55] and would provide potentially important additional information about the animal's social circumstance in an environment where vision cannot be employed (*Xenopus* reproduce at night in muddy ponds) [20].

The EOG technique we describe here may be useful to identify candidate social signaling molecules so that their behavioral effects can be evaluated. In other species, EOG has been a key tool in screening and identifying pheromones [27, 40, 41, 56]. The identification of such signals in *X. laevis* would be an important addition to the growing body of genetic, behavioral, physiological, and evolutionary knowledge about this species [17, 18].

## Acknowledgments

We thank Heather Eisthen for assistance and training with EOG techniques as well as helpful comments on a draft of this manuscript, Katie Darrah for her work piloting EOG recordings, and members of the Grass Lab 2017 for advice and support.

## Author Contributions

**Conceptualization:** Heather J. Rhodes.

**Formal analysis:** Heather J. Rhodes.

**Funding acquisition:** Heather J. Rhodes, Melanie Amo.

**Investigation:** Heather J. Rhodes, Melanie Amo.

**Methodology:** Heather J. Rhodes, Melanie Amo.

**Project administration:** Heather J. Rhodes.

**Writing – original draft:** Heather J. Rhodes.

**Writing – review & editing:** Melanie Amo.

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
