## [Decision Letter · Decision Letter 0]

15 Mar 2022

PONE-D-21-38952Do Xenopus laevis communicate through chemical signaling? The nose knows.PLOS ONE

Dear Dr. Rhodes,

Thank you for submitting your manuscript to PLOS ONE (and sorry for the delay). After careful consideration, we feel that it has merit but does not fully meet PLOS ONE’s publication criteria as it currently stands. Therefore, we invite you to submit a revised version of the manuscript that addresses the points raised during the review process. Please consider and respond to the reviewers' specific comments.

We look forward to receiving your revised manuscript.

Kind regards,

Michael Klymkowsky, Ph.D.

Academic Editor

PLOS ONE

Journal Requirements:

“We thank Heather Eisthen for assistance and training with EOG techniques as well as helpful comments on a draft of this manuscript, Katie Darrah for her work piloting EOG recordings, and members of the Grass Lab 2017 for advice and support. We also acknowledge and thank the following funding sources: The Grass Foundation, Denison University, the R.C. Good Faculty Fellowship, and the Helen L. Yeakel Summer Research Fund.”

“Funding was provided by The Grass Foundation https://grassfoundation.org/ (HJR), and Denison University https://denison.edu/, including the R.C. Good Faculty Fellowship at Denison University (HJR), and the Helen L. Yeakel Summer Research Fund at Denison University (MA). The funders had no role in study design, data collection and analysis, decision to publish, or preparation of the manuscript.”

Reviewers' comments:

Reviewer's Responses to Questions

**Comments to the Author**

1. Is the manuscript technically sound, and do the data support the conclusions?

Reviewer #1: Yes

Reviewer #2: Yes

2. Has the statistical analysis been performed appropriately and rigorously? 

Reviewer #1: Yes

Reviewer #2: Yes

3. Have the authors made all data underlying the findings in their manuscript fully available?

Reviewer #1: Yes

Reviewer #2: Yes

4. Is the manuscript presented in an intelligible fashion and written in standard English?

Reviewer #1: Yes

Reviewer #2: Yes

5. Review Comments to the Author

Reviewer #1: In their manuscript entitled "Do Xenopus laevis communicate through chemical signaling? The nose knows," Heather J. Rhodes and Melanie Amo developed an in situ electroolfactogram preparation and recorded olfactory responses in the adult water nose of Xenopus.

The study is very well conducted and straightforward. The logic of the experiments is fine, the methodology is sound, the statistical analysis is well done, and the authors provide convincing evidence for all their claims. I enjoyed reading the manuscript.

The results are extremely interesting for individuals working in the fields of chemical senses and (amphibian) olfaction in particular. To the best of my knowledge, this study is the first study that shows odorant responses recorded from the intact adult water nose of Xenopus laevis. The obtained results are, without a doubt, a significant advance in the relevant fields.

I find myself in the unusual position of not having major complaints. I only have a few very minor suggestions and points that the authors could consider taking into consideration.

Specific suggestions

I'm not too fond of questions as manuscript titles. Why don't you shortly state what you did in the title?

You could consider adding a figure that shows details about the in situ electroolfactogram preparation. This figure could also show the anatomy of the tripartite olfactory organ of adult Xenopus and give information about how you positioned the electrodes etc. Such a figure would be beneficial for many readers.

Lines 4-5: Here, you state that the adult water nose is well placed to receive information about conspecifics. Why do you think this is the case?

Line 27: What do you mean by "chemicals originating in the mouth"?

Line 32: I do not like the term "fishlike" class I olfactory receptors. I think there are more elegant and up-to-date ways to name/ describe these olfactory receptors.

Lines 83-89: What do you mean by "early animals"? Were the 27 male frogs already sexually mature? Also, you should add some more information about the male and female frogs from which you collected the stimuli (size, sexual maturity, etc.).

Line 99: ml/m? Do you mean ml/min?

Lines 114-121: You could be more precise when explaining how you placed the electrodes. But see also my suggestion to add a figure that better describes the recording of olfactory responses.

Lines 124-129: You could explain why L-methionine and L-alanine are suitable control stimuli for the adult water nose.

Line 228: Experiment 8? Why do you give this information? Is the number of the experiment important?

The discussion section is relatively short. Consider discussing some points in more detail. You could, for instance, speculate what morphological olfactory receptor neuron types and what olfactory receptor families could be responsible for the recorded odorant responses.

In the discussion section, you could compare the detection thresholds of the responses to amino acids obtained in your work with detection thresholds obtained in other studies using other methods. There is a paper (Breunig et al., 2010) where thresholds to amino acids, including L-methionine, of single Xenopus receptor neurons have been determined using the calcium imaging technique.

Line 323-325: Here, a reference for your statement is missing.

Reviewer #2: The authors describe olfactory responses in the water nose of Xenopus laevis to several bodily secretions of males and females using the electroolfactogram technique. Technically this appears to be a solid and carefully designed and performed study. The reported responses are an important step in understanding intraspecies olfactory communication in amphibians. However, the presentation of results and conclusions could be improved in several ways. Sometimes information is missing, misleading or incorrect. A detailed list follows:

Abstract, line 7

Please include information whether the explants are from males or females (this information is given for the source of the odors).

Line 13

Imprecise, this manuscript is not 'adding new layers' to understanding of 'vocal communication'.

Introduction, Line 26

Please explain the term 'choana' and explain the access of liquids to the VNO better.

Line 29

would be clearer to write 'through which water is actively circulated'

Line 61

Mention here which of the noses (water nose, air nose) are examined.

Line 62

Were female noses also examined?

Methods, line 92, 'double pithing the frog'

Please explain the procedure, as it is not generally known. Also 'euthanizing' means to kill, but since the heart rate is retained, this seems to be the wrong expression. 'Paralysed' might be a more correct term? Also please mention whether anesthesis is necessary/maintained during recording. You should also mention that the frog was killed after the experiment (I assume it was).

Line 95 'ice was placed over the frogs body'

Is that an approved method?

Line 119 'with expected timing'

Please specify the timing you expect here.

Line 135-135

Please mention the age range of animals pooled. Are there particular (hormonal) states to be considered?

Line 138

Which plastic? Different kinds might emit different contaminants.

Line 144-145

This is a good control, but not optimal as one could argue that the skin secretions might dissolve plastic components that pure saline could not. Maybe include a caveat here.

Line 155-157

Maybe rephrase for clarity in this way: After injection, the stimulus reached the olfactory epithelium after several seconds and somewhat diluted.

Line 159

Volume of injected NaCl?

Line 162-163

The sentence in parentheses should be a separate sentence and should be stated more directly: why you think the concentration at the olfactory epithelium could vary, and which direction do you expect it to vary.

Line 165-166

Give the volume of stimuli.

Line 176-178

Please mention whether results thus obtained are similar or different to those going from lowest to highest concentration.

Line 202-203 and 205-206

Please clarify: How can you subtract a signal outside of the recorded time range?

Results, line 245

This information (estimate of 5 fold dilution) should also be given in the respective Methods section. There the authors write only 'and some dilution of stimuli'.

Line 252

'may not be distinguishable'? Should read 'are not distinguishable' according to the criteria set by the authors.

Line 287-292

Please state here whether EOG responses to control stimuli reached significance according to your 2 sigma criteria.

Discussion, line 300-301

'estimated to be 2 µM' seems a rather strong statement. Please weaken according to the actual accuracy, which which you can estimate the dilution factor.

Line 302

What about female frogs? If not tested that should be stated explicitly here and before in Methods and Results.

Line 324

'held in amplexus' is not a generally known term. Please explain.

Line 330-331

One hopes. But is it really practical to use this EOG assay for some kind of biochemical fractionation? If the authors plan such studies that could be hinted at here.

Figures 3-5

Better to label these curves animal1, animal2 or individual1 etc, not 'exp1'. Also, this description could go inside the boxes, there is lots of empty space in the upper right corner.

Figure 3

Assuming that your 'exp1-6' are numbered chronologically there seems to be a very clear trend of decreasing z score with time (no such trend for the other curves). If true, please mention and explain (potential) causes for this trend. Is there a possibility it relates to season?

Figures 3-5

The x-axis lettering should mention the stimulus directly, including whether it is from male or female animals. Do not just write 'stimulus' there.

6. PLOS authors have the option to publish the peer review history of their article (what does this mean?). If published, this will include your full peer review and any attached files.

Reviewer #1: **Yes: **Ivan Manzini

Reviewer #2: No

---

## [Author Response · Author response to Decision Letter 0]

25 May 2022

Please see Response to Reviewers.docx. (the text of which is copied below, but lacks formatting here.)

Dear Editors and Reviewers,

Thank you for your thoughtful feedback on our manuscript, PONE-D-21-38952. We used your feedback to revise and improve our work and hope that our revisions are to your satisfaction. We feel the manuscript is stronger for this process.

Below, we offer a point-by-point response to all feedback. Journal and reviewer comments are in italics and our responses are in purple text. Please note that when we refer to line numbers in our responses, we give line numbers from the Track Changes version of the manuscript. 

We look forward to your response,

Heather J. Rhodes and Melanie Amo

Journal Requirements:

We have double checked style requirements and believe all is in order.

“We thank Heather Eisthen for assistance and training with EOG techniques as well as helpful comments on a draft of this manuscript, Katie Darrah for her work piloting EOG recordings, and members of the Grass Lab 2017 for advice and support. We also acknowledge and thank the following funding sources: The Grass Foundation, Denison University, the R.C. Good Faculty Fellowship, and the Helen L. Yeakel Summer Research Fund.”

“Funding was provided by The Grass Foundation https://grassfoundation.org/ (HJR), and Denison University https://denison.edu/, including the R.C. Good Faculty Fellowship at Denison University (HJR), and the Helen L. Yeakel Summer Research Fund at Denison University (MA). The funders had no role in study design, data collection and analysis, decision to publish, or preparation of the manuscript.”

We removed funding sources from the acknowledgement section. All funding sources are represented in the finding statement. 

No changes are needed. I am prepared to provide access upon acceptance.

Reference list is complete, correct, and to the best of our knowledge no sources have been retracted.

Reviewers' comments:

Reviewer #1: In their manuscript entitled "Do Xenopus laevis communicate through chemical signaling? The nose knows," Heather J. Rhodes and Melanie Amo developed an in situ electroolfactogram preparation and recorded olfactory responses in the adult water nose of Xenopus.

The study is very well conducted and straightforward. The logic of the experiments is fine, the methodology is sound, the statistical analysis is well done, and the authors provide convincing evidence for all their claims. I enjoyed reading the manuscript.

The results are extremely interesting for individuals working in the fields of chemical senses and (amphibian) olfaction in particular. To the best of my knowledge, this study is the first study that shows odorant responses recorded from the intact adult water nose of Xenopus laevis. The obtained results are, without a doubt, a significant advance in the relevant fields.

I find myself in the unusual position of not having major complaints. I only have a few very minor suggestions and points that the authors could consider taking into consideration.

Thank you. We are gratified that you found the study to be, for the most part, strong, valuable, and clearly reported. 

Specific suggestions

I'm not too fond of questions as manuscript titles. Why don't you shortly state what you did in the title?

I understand; question titles usually annoy me too. We altered the title to be a statement rather than a question.

You could consider adding a figure that shows details about the in situ electroolfactogram preparation. This figure could also show the anatomy of the tripartite olfactory organ of adult Xenopus and give information about how you positioned the electrodes etc. Such a figure would be beneficial for many readers.

We agree that such a figure would be helpful. Unfortunately, we do not have any high-quality photographs of our preparation. We developed schematics which we added as Fig 1. We hope these are useful.

Lines 4-5: Here, you state that the adult water nose is well placed to receive information about conspecifics. Why do you think this is the case?

We were alluding to the fact that olfaction is a good candidate for social communication. Chemical communication is common among aquatic animals. Xenopus secrete chemicals into the water through skin secretions and cloacal fluids and they sample the chemical composition of the water through the water nose. In case the specific words “well placed” were problematic or confusing, we have adjusted the wording of this sentence. We elaborate on these ideas in the third paragraph of the introduction (apx. Lines 52-62).

Line 27: What do you mean by "chemicals originating in the mouth"?

The source of stimuli to the adult Xenopus VNO is unclear, to the best of our knowledge. In some frogs, it seems that water passing through the nares may pass over the VNO, but given that Xenopus does not allow water to enter the principal cavity when submerged, instead directing water into the medial cavity, it seems unlikely that water from the nares would reach the VNO. Thus the choana or the nasolacrimal ducts seem likely pathways for stimuli to reach the VNO. We tried to clarify the language in the paper. 

Line 32: I do not like the term "fishlike" class I olfactory receptors. I think there are more elegant and up-to-date ways to name/ describe these olfactory receptors.

We revised this sentence to provide more specific and complete information.

Lines 83-89: What do you mean by "early animals"? Were the 27 male frogs already sexually mature? Also, you should add some more information about the male and female frogs from which you collected the stimuli (size, sexual maturity, etc.).

Please see the second paragraph in the methods which begins by stating that all animals were sexually mature adults. That includes all males used for physiology, as well as males and females used for stimulus acquisition. We did not obtain specific weights and lengths of animals used for stimulus acquisition, so we can provide no further information. “Early animals” simply meant those used for early experiments as we developed our procedures. We have clarified the language. 

Line 99: ml/m? Do you mean ml/min?

Yes – thanks for catching that!

Lines 114-121: You could be more precise when explaining how you placed the electrodes. But see also my suggestion to add a figure that better describes the recording of olfactory responses.

A bit of additional information was added, as well as the new Fig 1. We hope this provides the information you’re looking for. 

Lines 124-129: You could explain why L-methionine and L-alanine are suitable control stimuli for the adult water nose.

We added information and citations to support the use of amino acids as positive controls. They have been recognized as odorants Xenopus can detect in the past. Additionally, they effectively drive odorant responses in a wide range of animals, including other frogs, salamanders, fish… even aquatic invertebrates. 

Line 228: Experiment 8? Why do you give this information? Is the number of the experiment important?

This information was provided so that readers can see how the sample data in figure 3 (renumbered from Fig 2) fit with the summary data graphed in figures 4 and 5.(renumbered from Figs 3 and 4). Experiments on individual frogs are numbered so that a reader can examine how a single animal responds to multiple stimuli. We have moved the reference to “experiment 8” out of the figure title and to the end of the legend in the hopes that it will be less distracting to readers.

The discussion section is relatively short. Consider discussing some points in more detail. You could, for instance, speculate what morphological olfactory receptor neuron types and what olfactory receptor families could be responsible for the recorded odorant responses.

We have modestly expanded the discussion in our revisions, but we don’t feel a compelling need to speculate on receptor types, as we’re not sure we have new insights to offer. The candidate list for molecules in the skin secretions and cloacal fluids that could be triggering these responses is still long and open ended. We appreciate the suggestion, but think this is a topic best left for future work.

In the discussion section, you could compare the detection thresholds of the responses to amino acids obtained in your work with detection thresholds obtained in other studies using other methods. There is a paper (Breunig et al., 2010) where thresholds to amino acids, including L-methionine, of single Xenopus receptor neurons have been determined using the calcium imaging technique.

Thanks for pointing this paper out! We have included it in the discussion.

Line 323-325: Here, a reference for your statement is missing.

We added citations that further support this idea.

Reviewer #2: The authors describe olfactory responses in the water nose of Xenopus laevis to several bodily secretions of males and females using the electroolfactogram technique. Technically this appears to be a solid and carefully designed and performed study. The reported responses are an important step in understanding intraspecies olfactory communication in amphibians. However, the presentation of results and conclusions could be improved in several ways. Sometimes information is missing, misleading or incorrect. A detailed list follows:

Thank you. We are sorry if points were unclear but stand by our work as correct; we will attempt to clear up any misunderstandings. 

Abstract, line 7

Please include information whether the explants are from males or females (this information is given for the source of the odors).

There are no “explants.” The preparation is in situ in that the olfactory epithelium is intact and in place, but the frog has been euthanized, so it is not in vivo. All EOG recordings were made from adult males, as is specified on lines 7-8. Language in the methods has been revised and a new Fig 1 has been added to clarify the nature of the preparation; we also state clearly that no females were used for electrophysiology.

Line 13

Imprecise, this manuscript is not 'adding new layers' to understanding of 'vocal communication'.

The intention was not to suggest that this paper alone adds to our understanding of vocal communication, but that continued research (as described in the previous sentence, lines 10-12) will. We have clarified the language.

Introduction, Line 26

Please explain the term 'choana' and explain the access of liquids to the VNO better.

The choana is the opening that connects the principal cavity of the nose with the oral cavity. A parenthetical has been added to better explain this specialized term, as well as a new Fig 1 has been added to provide a visual reference. The choana is part of the path air follows to the lungs. This opening may also provide a path for non-volatile chemicals to reach the VNO, which sits in the bottom of the principal cavity, adjacent to the choana. It is also possible that the nasolacrimal duct provides a path for fluid and stimuli to reach the VNO. In some frogs, it seems that water passing through the nares may pass over the VNO, but given that Xenopus does not allow water to enter the principal cavity when submerged, instead directing water into the medial cavity, it seems unlikely that water from the nares would reach the VNO. This anatomy is complex, we know. We hop the new figure will help clarify. 

Line 29

would be clearer to write 'through which water is actively circulated'

Change made – thank you.

Line 61

Mention here which of the noses (water nose, air nose) are examined.

Done. It was the water nose – thank you.

Line 62

Were female noses also examined?

No. This was restated in the methods to clarify.

Methods, line 92, 'double pithing the frog'

Please explain the procedure, as it is not generally known. Also 'euthanizing' means to kill, but since the heart rate is retained, this seems to be the wrong expression. 'Paralysed' might be a more correct term? Also please mention whether anesthesis is necessary/maintained during recording. You should also mention that the frog was killed after the experiment (I assume it was).

Pithing is an accepted form of euthanasia for many animals, including anuran amphibians. Double pithing a frog involves the destruction of all central nervous system structures with a probe inserted through the foramen magnum into the skull, and then into the spinal cord. The probe is moved in a circular motion to destroy all brain and spinal cord structures and nerve connections. We have added a citation from the American Veterinary Medical Association to support this form of euthanasia as well as a short explanation of pithing in the subsequent sentence. We also attempted to clarify the final sentence in the paragraph. 

We are happy to provide further information, although it feels like it may take the paper off track to explain amphibian biology at length. Once the frog is pithed, it is brain dead and the animal no longer breathes. But the tissues of these animals are fairly hypoxia tolerant and some carbon dioxide can be exchanged through the skin, preventing hypercapnia. The myogenic heart may continue to beat for a couple of hours after brain death. It also continues to beat for hours after decapitation and can even continue to beat for a considerable time if the heart is fully removed from the body. Thus, the heart is not a good indicator of death in the frog. Brain death, from the total destruction of the central nervous system, is a better criterion. After pithing, the frog is dead, even if particular cells and tissues remain functional.

Line 95 'ice was placed over the frogs body'

Is that an approved method?

All procedures were approved by IACUC as written. Ice is generally not allowed in the place of sedation or anesthesia, but here anesthesia was achieved by an injectable sodium channel blocker (MS222) first. Ice was utilized to decrease metabolic rate in the tissues with the intent of extending the time over which the preparation would remain viable. 

Line 119 'with expected timing'

Please specify the timing you expect here.

A clarification was added.

Line 135-135

Please mention the age range of animals pooled. Are there particular (hormonal) states to be considered?

They were all adults, sexually mature, and were not part of any current breeding programs, but beyond that we do not have any specific information. We added a sentence to clarify what we do and don’t know.

Line 138

Which plastic? Different kinds might emit different contaminants.

Sandwich bags are made of low-density polyethylene. We have added this information to the paper.

Line 144-145

This is a good control, but not optimal as one could argue that the skin secretions might dissolve plastic components that pure saline could not. Maybe include a caveat here.

This is possible, but it seems far more likely that the many water-soluble compounds in the skin secretions are what is being detected by the animal. We think this was an appropriate control, clearly explained. We are satisfied that a reader, such as yourself, may wonder about the possibility of dissolved plastics, but we don’t think a caveat is needed. 

Line 155-157

Maybe rephrase for clarity in this way: After injection, the stimulus reached the olfactory epithelium after several seconds and somewhat diluted.

Thanks for the suggestion – it was implemented.

Line 159

Volume of injected NaCl?

50µl – the same as a stimulus. This was added to the text.

Line 162-163

The sentence in parentheses should be a separate sentence and should be stated more directly: why you think the concentration at the olfactory epithelium could vary, and which direction do you expect it to vary.

We really can’t say. We were collecting drops of water from the perfusion system, then testing that later on an osmometer. That could have acted like a smoothing function on the concentration over time, removing peaks or valleys. In the preparation, perfusate is allowed to flow freely out of the nasal cavity, but there could be some slight pooling or mixing of liquid on the epithelium that could further alter concentration on the surface over time. The EOG responses themselves also reflect relatively slow and long-lasting receptor potentials, suggesting integration over time in the olfactory epithelium. Thus, even if the concentration delivered is higher or lower for short periods of time (in ways we were unable to measure) the olfactory epithelium itself could act as a smoothing function. Our approach was to get a sense of the timing and dilution and report that transparently to the best of our ability. We have changed these sentences to ensure the information is clear without offering supposition.

Line 165-166

Give the volume of stimuli.

This is reported in the first line of the paragraph that starts “To deliver a stimulus…” which is now line 175. It was also added on line 181 for further clarity.

Line 176-178

Please mention whether results thus obtained are similar or different to those going from lowest to highest concentration.

This has been added to the manuscript. We saw no differences.

Line 202-203 and 205-206

Please clarify: How can you subtract a signal outside of the recorded time range?

Recording was continuous for the entire length of a stimulus block (so a single recording file might be 40+ minutes long). Note on line 127-128 it says that data was “continuously recorded.”

Results, line 245

This information (estimate of 5 fold dilution) should also be given in the respective Methods section. There the authors write only 'and some dilution of stimuli'.

The 5-fold (or 1:5) dilution is described in the methods section. See specifically lines 182-190.

Line 252

'may not be distinguishable'? Should read 'are not distinguishable' according to the criteria set by the authors.

We do not claim to know whether stimuli are or are not detectable by the animals. Behavioral experiments would be needed to determine that. We carefully stated at the end of the methods (now lines 247-249) that “z ≥ 2 was set to represent responses that are likely detectable as different from control, since such responses would be greater in amplitude than 97.7% of control responses.” Saying stimuli beyond that threshold are “likely detectable” does not mean stimuli below that threshold are not detectable. Thus, the criteria we set are consistent with the language in the figure legend: “light gray horizontal band from z = -2 to z = 2 indicates responses that may not be distinguishable from saline control.”

Line 287-292

Please state here whether EOG responses to control stimuli reached significance according to your 2 sigma criteria.

It was the standard deviation of these control stimuli that was used in the z-score calculation. This was stated on lines 240-246 of the methods, but we clarified in the results on lines 270-272.

Discussion, line 300-301

'estimated to be 2 µM' seems a rather strong statement. Please weaken according to the actual accuracy, which which you can estimate the dilution factor.

This statement has been modified accordingly.

Line 302

What about female frogs? If not tested that should be stated explicitly here and before in Methods and Results.

A statement was added to the methods to make it clear that only males were used for EOG experiments. In both the introduction (lines 38-61) and the discussion (lines 345 to the end) we provide a description of the various vocal and clasping behaviors that males engage in, and use differentially in the presence of other males or females. Thus, we had reason to look at male EOG responses because we are interested in what sensory cues males use to select appropriate behaviors. 

Line 324

'held in amplexus' is not a generally known term. Please explain.

This is the reproductive clasping position used by anuran amphibians. In the case of Xenopus, the male’s head is on the female’s back, with his forelimbs clasped around her abdomen. In this position, the female lays eggs and the male releases sperm for external fertilization. Males also hold other males in amplexus, likely as an alternative reproductive tactic to gain proximity to a female and engage in sperm competition. We added a brief explanation of the term and citations to support.

Line 330-331

One hopes. But is it really practical to use this EOG assay for some kind of biochemical fractionation? If the authors plan such studies that could be hinted at here.

It has indeed been done in other species and we hope it will be in Xenopus, by us or others. I have elaborated on these ideas and added citations. 

Figures 3-5

Better to label these curves animal1, animal2 or individual1 etc, not 'exp1'. Also, this description could go inside the boxes, there is lots of empty space in the upper right corner.

We have changed the labels to “Frog#” instead of “exp#” and moved the keys into the space of the graphs. 

Figure 3

Assuming that your 'exp1-6' are numbered chronologically there seems to be a very clear trend of decreasing z score with time (no such trend for the other curves). If true, please mention and explain (potential) causes for this trend. Is there a possibility it relates to season?

We think this was just due to random chance. As you note, there is not a relationship between z-scores and experiment order for other stimuli. All of the data reported here was collected within a one-month time period mid-July to mid-August and animals were housed under tightly controlled conditions, making seasonal change unlikely. 

Figures 3-5

The x-axis lettering should mention the stimulus directly, including whether it is from male or female animals. Do not just write 'stimulus' there.

We have revised the titles and labels on the x-axes of all graphs.

---

## [Decision Letter · Decision Letter 1]

18 Jul 2022

PONE-D-21-38952R1Electrophysiological responses to conspecific odorants in Xenopus laevis show potential for chemical signaling.PLOS ONE

Dear Dr. Rhodes,

Thank you for submitting your manuscript to PLOS ONE. After careful consideration, we feel that it has merit but does not fully meet PLOS ONE’s publication criteria as it currently stands. Therefore, we invite you to submit a revised version of the manuscript that addresses the points raised during the review process.

Please address review #2's minor comments and return the revised manuscript, no further review will be necessary

We look forward to receiving your revised manuscript.

Kind regards,

Michael Klymkowsky, Ph.D.

Academic Editor

PLOS ONE

Journal Requirements:

Reviewers' comments:

Reviewer's Responses to Questions

**Comments to the Author**

1. If the authors have adequately addressed your comments raised in a previous round of review and you feel that this manuscript is now acceptable for publication, you may indicate that here to bypass the “Comments to the Author” section, enter your conflict of interest statement in the “Confidential to Editor” section, and submit your "Accept" recommendation.

Reviewer #1: All comments have been addressed

Reviewer #2: (No Response)

2. Is the manuscript technically sound, and do the data support the conclusions?

Reviewer #1: Yes

Reviewer #2: Yes

3. Has the statistical analysis been performed appropriately and rigorously? 

Reviewer #1: Yes

Reviewer #2: Yes

4. Have the authors made all data underlying the findings in their manuscript fully available?

Reviewer #1: Yes

Reviewer #2: Yes

5. Is the manuscript presented in an intelligible fashion and written in standard English?

Reviewer #1: Yes

Reviewer #2: Yes

6. Review Comments to the Author

Reviewer #1: (No Response)

Reviewer #2: The manuscript has much improved in the revision, in particular the readability for a larger audience. The addition of a schematic figure is welcome.

Minor points:

Figure 1: the opening of the nose (naris) is a white oval in panel A, but a black dot in panel B. Better use the white oval also in panel B, this has the advantage of not having to distinguish between small and large black dots.

Figure 1: The drawing looks like the VNO is accessed from the MC. Can you make the drawing so that VNO looks connected to PC, not to MC?

Line 36 „ciliated and microvillous receptor neurons expressing class I OR1, OR2, and V1R receptors“

I think you mean class I and class II ORs, not ' class I OR1, OR2'.

Line 135-136, „a characteristic negative deflection lasting 2-3 seconds with expected timing based on the perfusion and stimulus deliver system described below)“

Unclear. Better say: „a characteristic negative deflection lasting 2-3 seconds, which is the expected timing based on the perfusion and stimulus deliver system described below)“

7. PLOS authors have the option to publish the peer review history of their article (what does this mean?). If published, this will include your full peer review and any attached files.

Reviewer #1: No

Reviewer #2: No

---

## [Author Response · Author response to Decision Letter 1]

31 Jul 2022

Minor modifications were made to figure one in accordance with the suggestions of reviewer 2. Reviewer 2 also asked for two minor modifications to the text; in both cases the reviewer's suggestions would have changed the meaning of the text and made it inaccurate. Instead, we attempted to clarify the text in light of the reviewer's misunderstanding. We hope the editor is happy with these modifications.

---

## [Editor Report · Decision Letter 2]

2 Aug 2022

Electrophysiological responses to conspecific odorants in Xenopus laevis show potential for chemical signaling.

PONE-D-21-38952R2

Dear Dr. Rhodes,

We’re pleased to inform you that your manuscript has been judged scientifically suitable for publication and will be formally accepted for publication once it meets all outstanding technical requirements.

Kind regards,

Michael Klymkowsky, Ph.D.

Academic Editor

PLOS ONE
---

## [Editor Report · Acceptance letter]

26 Aug 2022

PONE-D-21-38952R2 

Electrophysiological responses to conspecific odorants in *Xenopus laevis* show potential for chemical signaling. 

Dear Dr. Rhodes:

I'm pleased to inform you that your manuscript has been deemed suitable for publication in PLOS ONE. Congratulations! Your manuscript is now with our production department. 

Kind regards, 

on behalf of

Dr. Michael Klymkowsky 

Academic Editor

PLOS ONE